# Effect of Covid-19 on maternal and child health services utilization in Ghana. Evidence from the National Health Insurance Scheme (NHIS)

**Yaw Nyarko Opoku-Boateng**[1,2], **Emmanuel Opoku-Asante**[1], **Mylene Lagarde**[2], **Edward Nketiah-Amponsah**[3] *

1 Claims Directorate, National Health Insurance Authority, Accra, Ghana, 2 Department of Health Policy, London School of Economics and Political Science, London, United Kingdom, 3 Department of Economics, University of Ghana, Legon, Accra, Ghana

* enamponsah@ug.edu.gh

**Data Availability Statement:** All relevant data are within the paper and Supporting Information files.

## Abstract

### Introduction

Covid-19 has had devastating effect on health systems and health utilization globally. Maternal and newborn care were adversely affected but little or nothing is known about the impact it has caused to it. This study seeks to determine the effect of Covid-19 on healthcare utilization with specifics on Antenatal, Postnatal, Deliveries and Out-patient attendance.

### Methods

The study uses secondary data obtained from the four (4) Claims Processing Centres of the National Health Insurance Authority. Through the use of convenient sampling, a total of 502 facilities were selected for inclusion in the research. The study used a longitudinal claims submitted from a cross-section of health facilities namely Community-Based Health Planning and Services, Maternity Homes, Health Centers, Clinics, Primary, Secondary, and Tertiary Hospitals for Antenatal, Postnatal, Out-patient consultations and Delivery attendances from January 2018 to December 2021. Data before and during the Covid-19 pandemic were compared. Segmented regression analysis as an interrupted time series analysis was employed to assess the effect of the pandemic on utilization of services.

### Results

The results indicate that Covid-19 had a significant impact on healthcare utilization in Ghana. Month-on-month, antenatal and out-patient utilization decreased by 21,948.21 and 151,342.40, respectively. Postnatal and delivery services saw an insignificant monthly increase of 37.76 and 1,795.83 from the onset of the covid-19 pandemic and the introduction of movement restrictions. This decline was observed across all care levels, except for Community-Based Health Planning and Services, which showed a slight increase. Also, the results indicate projected average misses of scheduled antenatal, postnatal, out-patient reviews, and deliveries at 21,037.75, 6,428.23, 141,395.30 and 4,745.63 patients respectively.

**Funding:** The author(s) received no specific funding for this work.

**Competing interests:** The authors have declared that no competing interests exist.

## Conclusion

The study reveals that Covid-19 led to a decrease in utilization of healthcare which affected pregnant women and newborn care as well. It was evident from the results that community-based healthcare is more resilient and efficient in delivering healthcare amidst the pandemic. In our quest to achieve Universal Health Coverage by 2030, Ghana's health system should improve on the community-based healthcare system and include technology in its healthcare delivery for the people.

## 1. Introduction

The COVID-19 pandemic has caused serious disruptions to healthcare systems globally. Ghana was among the early countries to impose restrictions on movement in some parts of the country following the confirmation of the first two cases of COVID-19 on March 12, 2020. Though emergency and essential services, including workers providing these services were excluded from the restrictions in movement by the government, the lockdown greatly affected healthcare delivery. Some maternity units and Neonatal Intensive Care Units (NICUs) were temporarily closed, as a result of infections among health professionals which led to deaths of patients [1,2]. The fear of contracting the virus prevented the sick who needed care from going to the hospital.

While lockdown measures were taken to minimize the immediate health effects of the pandemic, there were concerns about the indirect effects, especially on vulnerable groups such as pregnant women and children. A study on the 2014 Ebola outbreak in West Africa have shown that the indirect effects of a health crisis can be more severe than the outbreak itself [3]. The study further found that the 2014 Ebola outbreak led to reduction in maternal delivery care in the affected areas by about 80% and admissions for malaria among the under five years reduced by 40% nationwide.

Studies that have examined the indirect effects of the COVID-19 pandemic suggests a near universal reduction in healthcare utilization across both low- and high-income countries [4–6]. However, evidence on the impact of the COVID-19 pandemic on healthcare utilization for pregnant women and newborns (neonatal) care in Ghana (and from sub-Saharan Africa) is scarce. Some evidence from Uganda suggest that antenatal services utilization during the COVID-19 period were lower than pre-COVID levels [4].

Optimal healthcare delivery has been a socio-economic challenge in Ghana because of poor distribution of health workforce and the limited available resources which has been worsened by the emergence of COVID-19. Among the most affected at these difficult times are the pregnant women and children. In Ghana, maternal mortality is excessively high at 319 deaths per 100,000 live births due to poor health seeking behaviour and it is feared that, Covid-19 pandemic will have a significant effect on the already existing weak health system [7].

In Ghana, the health system makes adequate provision for maternal-child care through free maternal-child care in National Health Insurance Scheme (NHIS) credentialed facilities but not without challenges. Healthcare seekers sometimes travel longer distances to access healthcare facilities. Now with travel and gathering restrictions, coupled with health facilities with insufficient infection prevention supplies and ineffective infection control methods, as well as disrupted community health worker routines, threaten to exacerbate women's health problems. Again, pandemics of this nature have proven to have a significant effect on health systems, with some indirect effects resulting not only in excess mortality, health service utilization

but also range of morbidities due to lack of preventive care and delay in diagnosis and disruptions to management of chronic conditions [5].

With a 9.6 positivity rate of Covid-19 infection [7], a complete lockdown measures were put in place by the government and the Ghana Health Service (GHS) in the Greater Accra and Greater Kumasi regions to help control the spread of the disease. The lockdown affected free movement of persons and hence limited activities in the said regions and the country at large since these two regions are the most productive regions in the country.

Studies on the direct and indirect effects of the COVID-19 pandemic on healthcare utilization and health outcomes abound [4,6,8–10]. An important focus of many studies has been the impact of the pandemic on vulnerable groups such as pregnant women and children [4,8,11]. For example, Singh et al. [8] found significantly large decreases in antenatal care services (22.91%) and immunization services (20%). Evidence from the UK and USA showed that pre-term deliveries among women with COVID-19 were higher than uninfected women [10]. There is evidence that suggest that pregnant women are more vulnerable to COVID-19, and that there is increased prevalence of preterm deliveries [5,12]. No published research has investigated the impact of covid-19 on healthcare utilization for pregnant women and newborn (neonatal) care in Ghana, including the study setting, to the best of the authors' knowledge. To bridge this gap, this study aims to explore the impact of covid-19 on healthcare utilization for maternal and new born care in Ghana. More specifically, we first estimate the total daily number of Out-patient (OPD) attendance before Covid-19 (2018 and 2019) and within Covid-19 pandemic period (2020 and 2021) to know the effect on utilization amidst covid-19 pandemic. Secondly, we estimate the daily Ante Natal Care (ANC) and Post Natal Care (PNC) and new-born care visits at the healthcare facility before and after the emergence of Covid-19 pandemic to know the effect on utilization by covid-19 pandemic. Finally, we estimate the daily number of Deliveries (DEL) conducted in the healthcare facilities before and after the emergence of Covid-19 pandemic to know the effect on utilization by covid-19 pandemic.

## 2. Methodology

To better understand the effect of hospital attendance in Ghana during the Covid-19 pandemic, the research follows a quantitative design. The primary motivation for adapting this research design lies in the research gap (*Utilization of health care during Covid-19 using Ghana's National Health Insurance data*). Also, quantitative design allows for classification of relationships among different variables in the dataset, making it possible to check for significance of the results.

### 2.1 Data collection method

The study utilizes secondary data obtained from the four (4) Claims Processing Centres (CPCs) of the National Health Insurance Authority (NHIA). The data contained health insurance claims submitted from a cross-section of health facilities (as shown in Tables 1–3 in S1 Appendix) namely Community-Based Health Planning and Services (CHPS), Maternity Homes, Health Centers, Clinics, Primary, Secondary, and Tertiary Hospitals for ANC, PNC, OPD and DEL attendances from January 2018 to December 2021. These health facilities were of different ownership types including public, private and mission. The choice of the specialties aligns with the study's primary focus on maternal and child health, ensuring a comprehensive analysis. Furthermore, inclusion criteria prioritized data availability and completeness for ANC, PNC, DEL, and OPD services, contributing to a comprehensive analysis of healthcare utilization in the context of maternal and child health. On the other hand, consultations that fell outside the specified specialties (ANC, PNC, DEL, and OPD) were not included in the

research sample. The reason for this omission was to ensure a specific emphasis on healthcare services directly related to maternal and child health, in line with the major goals of the study. In addition, consultations that lacked complete or adequate data were eliminated to maintain the integrity of the analysis. The intentional omission of cases with missing data was done to reduce potential biases and improve the accuracy of the results.

To ensure that healthcare utilization is not biased by delayed reimbursement, we collected claims data from health facilities that had no outstanding claims submissions from 2018 to 2021 as of the time of data collection. Convenient sampling, which is a non-probabilistic sampling design was used to select these health facilities. In all, a total of 502 distinct health facilities were identified across the sixteen (16) regions of Ghana. A total of over 37 million claims data from the 502 distinct health facilities were obtained across the various CPCs for the purpose of the study. In order not to compromise on patients' privacy, only *date of attendance*, *facility name*, *region*, *district*, *and volume of claims* for the specialties of interest were collected. A summary of the total hospital attendances by specialty over the period are shown in *Table* 1 below. The total month-on-month detailed attendances are presented the in *Table* 2.

## 2.2 Ethics approval

The study uses secondary data collated from the Claims Processing Centre of the National Health Insurance Authority. Ethical approval is not required for the use such secondary data in Ghana. This quantitative data mainly on antenatal and postnatal care attendance, outpatient consultations and delivery attendances from January 2018 to December 2021 were aggregated and did not involve interviews, responses or identification with any human participant. The data is freely available to the public upon request from the National Health Insurance Authority. The lead author is the Deputy Director of the NHIA Claims Processing Centre, the unit responsible for the generation of the claims data. Nevertheless, permission for the use of the data was sought from the Chief Executive Officer of the National Health Insurance Authority (see supporting information).

## 2.3 Data analysis

Data before and during the Covid-19 pandemic were compared. The Statistical software package *Stata-16* was used to conduct the analysis. To be able to control trends, segmented regression analysis as an interrupted time series analysis was employed for this study to assess the effect of the pandemic on utilization of services due to the longitudinal nature of the data as elaborated by Ramsay et al. [13], Wagner et al. [14], Lagarde [15]. According to Habib et al. [16], segmented regressions are appropriate to analyses time series data that require the study of the effect of an intervention.

The data that were obtained from the CPCs were further put into an appropriate layout for the purpose of the analysis. The initial data was made up of 733,422 observations, thus each of the 502 facilities had a minimum of 365 observations for each year *(2018,2019 and 2021 had*

**Table 1. Total hospital attendances by specialty.**

| Year | ANC | PNC | OPD | DEL | Total |
|---|---|---|---|---|---|
| 2018 | 1,335,782 | 57,014 | 6,513,418 | 256,346 | 8,162,560 |
| 2019 | 1,386,769 | 133,859 | 7,647,578 | 276,635 | 9,444,841 |
| 2020 | 1,358,029 | 188,293 | 6,666,760 | 304,683 | 8,517,765 |
| 2021 | 1,752,044 | 240,211 | 8,978,954 | 371,961 | 11,343,170 |
| **Total** | **5,832,624** | **619,377** | **29,806,710** | **1,209,625** | **37,468,336** |

**Table 2. Variables used for the analysis.**

| Month | ANC | PNC | OPD | DEL | Time | Pandemic | Postslope |
|---|---|---|---|---|---|---|---|
| Jan-18 | 113342 | 5215 | 531006 | 19656 | 1 | 0 | 0 |
| Feb-18 | 98625 | 3798 | 504961 | 17374 | 2 | 0 | 0 |
| Mar-18 | 109166 | 5403 | 497634 | 20684 | 3 | 0 | 0 |
| Apr-18 | 130299 | 5427 | 585148 | 22325 | 4 | 0 | 0 |
| May-18 | 113672 | 5659 | 507730 | 24823 | 5 | 0 | 0 |
| Jun-18 | 105673 | 4939 | 519780 | 22368 | 6 | 0 | 0 |
| Jul-18 | 104324 | 4885 | 585586 | 21296 | 7 | 0 | 0 |
| Aug-18 | 120315 | 3962 | 561011 | 18611 | 8 | 0 | 0 |
| Sep-18 | 99376 | 4700 | 494275 | 22460 | 9 | 0 | 0 |
| Oct-18 | 112223 | 5204 | 608321 | 25403 | 10 | 0 | 0 |
| Nov-18 | 106699 | 4183 | 551622 | 20503 | 11 | 0 | 0 |
| Dec-18 | 122068 | 3639 | 566344 | 20843 | 12 | 0 | 0 |
| Jan-19 | 102500 | 2892 | 503209 | 19690 | 13 | 0 | 0 |
| Feb-19 | 105032 | 2760 | 514945 | 18593 | 14 | 0 | 0 |
| Mar-19 | 105911 | 3016 | 544133 | 22328 | 15 | 0 | 0 |
| Apr-19 | 117178 | 11400 | 577873 | 23963 | 16 | 0 | 0 |
| May-19 | 123149 | 13102 | 627949 | 25921 | 17 | 0 | 0 |
| Jun-19 | 111056 | 13014 | 655635 | 23908 | 18 | 0 | 0 |
| Jul-19 | 121971 | 14267 | 749052 | 22260 | 19 | 0 | 0 |
| Aug-19 | 118094 | 13515 | 662619 | 21389 | 20 | 0 | 0 |
| Sep-19 | 118990 | 14264 | 626890 | 24167 | 21 | 0 | 0 |
| Oct-19 | 130844 | 16595 | 814014 | 26991 | 22 | 0 | 0 |
| Nov-19 | 120253 | 15988 | 756905 | 23980 | 23 | 0 | 0 |
| Dec-19 | 111791 | 13046 | 614354 | 23445 | 24 | 0 | 0 |
| Jan-20 | 129660 | 16519 | 660756 | 23097 | 25 | 0 | 0 |
| Feb-20 | 121429 | 14554 | 644163 | 21954 | 26 | 0 | 0 |
| Mar-20 | 122637 | 15666 | 586295 | 27128 | 27 | 1 | 1 |
| Apr-20 | 107969 | 14406 | 424562 | 28023 | 28 | 1 | 2 |
| May-20 | 98909 | 14061 | 439038 | 29705 | 29 | 1 | 3 |
| Jun-20 | 107686 | 14481 | 484547 | 25790 | 30 | 1 | 4 |
| Jul-20 | 104526 | 14877 | 529624 | 23804 | 31 | 1 | 5 |
| Aug-20 | 101671 | 14358 | 505323 | 22778 | 32 | 1 | 6 |
| Sep-20 | 106229 | 14678 | 492536 | 23583 | 33 | 1 | 7 |
| Oct-20 | 116533 | 18720 | 606067 | 27814 | 34 | 1 | 8 |
| Nov-20 | 119614 | 18416 | 658976 | 26822 | 35 | 1 | 9 |
| Dec-20 | 121166 | 17557 | 634873 | 24185 | 36 | 1 | 10 |
| Jan-21 | 118301 | 17099 | 577978 | 24364 | 37 | 1 | 11 |
| Feb-21 | 115605 | 15136 | 550281 | 23013 | 38 | 1 | 12 |
| Mar-21 | 135208 | 18507 | 576158 | 28190 | 39 | 1 | 13 |
| Apr-21 | 120280 | 19347 | 510299 | 30613 | 40 | 1 | 14 |
| May-21 | 126254 | 19303 | 574638 | 33213 | 41 | 1 | 15 |
| Jun-21 | 129932 | 19404 | 659076 | 30027 | 42 | 1 | 16 |
| Jul-21 | 129895 | 18741 | 770705 | 28129 | 43 | 1 | 17 |
| Aug-21 | 156760 | 20439 | 882701 | 30644 | 44 | 1 | 18 |
| Sep-21 | 173997 | 22159 | 907675 | 35021 | 45 | 1 | 19 |
| Oct-21 | 178307 | 23927 | 1011954 | 37637 | 46 | 1 | 20 |
| Nov-21 | 190881 | 23666 | 1003526 | 36329 | 47 | 1 | 21 |
| Dec-21 | 176624 | 22483 | 953963 | 34781 | 48 | 1 | 22 |

**Table 3. Descriptive statistics of the dataset covid-19 (precovid-19).**

| Variable | Mean | Std. Dev. | Min | Max |
|---|---|---|---|---|
| ANC | 121,513.00 | 21,037.75 | 98,625 | 190,881 |
| DEL | 25,200.52 | 4,745.63 | 17,374 | 37,637 |
| PNC | 12,903.69 | 6,428.23 | 2,760 | 23,927 |
| OPD | 620,973.10 | 141,395.30 | 424,562 | 1,011,954 |
| Total | 780,590.31 | 168,049.60 | 574,960 | 1,254,402 |

*365 days whiles 2020 had 366 days* since it was a leap year, totaling 1,461 days for each facility over the 4-year period with an average of 30 days each month). The various attendances were put into various months of the year as shown in Table 3. A total of 502 facilities without outstanding claims submissions over the period were used for the study. Over 30 million sampled claims had been submitted by these facilities over the 4-year period for the specialties of interest (ANC, PNC, OPD and DEL).

In summary, the data covers a total of 48 months as indicated by Time in Table 3. To be able to carry out the segmented regression, we created additional independent variables namely time, pandemic and postslope. Time represents the beginning to the end of the observation period. Time is a continuous variable and relates to each month over the period. Pandemic is the intervention period, thus the period before Covid (precovid-19) and during Covid-19. Precovid-19 spans from January 2018 to February 2020 with March 2020 to December 2021classified as Covid-19 period since the first two cases in Ghana were confirmed on 12 March 2020. Precovid-19 was coded as 0 and 1 for Covid-19. Finally, postslope was coded 0 to the point when the pandemic (covid19) was announced in Ghana and thereafter starts at 1 and relics as a continuous variable to the last end of the period.

Time, pandemic and postslope were used as independent variables in the quest to find out if Covid-19 had affected hospital attendance in any way. Again, to critically analyze and estimate the quantitative effect of the pandemic on the various specialties of interest; ANC, PNC, OPD and DEL were also used as the dependent variables in each case.

A linear regression was employed to estimate the effect of covid-19 on health care utilization. The main equation is stated as follows:

$$Y_t = \beta_0 + \beta_1 * time_t + \beta_2 * pandemic_t + \beta_3 * postslope_t + \varepsilon_t$$

The outcome (dependent) variable $Y_t$ *is a measure of number of hospital attendance per month*, $\beta_0$ estimates the *of number of hospital attendance* at time zero; $\beta_1$ estimates the *changes in hospital attendance* for each month before the pandemic; $\beta_2$ estimates the *changes in hospital attendance* during the pandemic and $\beta_3$ estimates the *growth rate in outcome during the pandemic*. The error term $\varepsilon_t$ at time t represents the random variability that is not explained by the linear regressions model.

## 3. Results

Table 3 reports the descriptive statistics of the dataset for the periods. The mean ANC attendance over the period was 121,513 with a standard deviation of 21,037.75. This implies that on average 21,037.75 patients are likely to skip a scheduled ANC. The mean for PNC was 12,903.69, standard deviation of 6,428.23. The mean for DEL was 25,200.52 with standard deviation in 4,745.63. On average, 141,395.30 patients are likely to miss their scheduled OPD reviews as indicated by the standard deviation. The mean monthly OPD consultation was 620,973.10.

**Table 4. Results of segmented linear regression model, OPD attendances.**

| Independent Variables | Coefficient | Std. Error | t | P-value |
|---|---|---|---|---|
| Constant | 526881 | 72586.97 | 7.26 | 0.00 |
| Baseline trend | 3448.22 | 4363.834 | 0.79 | 0.434 |
| Change during covid | -151342 | 68426.69 | -2.21 | 0.032 |
| Change in utilization | 15615.94 | 7899.996 | 1.98 | 0.054 |

With regard to the quantitative analysis, we estimated the effect of Covid-19 pandemic on health care utilization across various health care levels in Ghana using claims data obtained from the CPCs of the NHIA from January 2018 to December 2021.

Table 4 presents estimate of the effect of the pandemic on OPD, and it is evident that at the start of the period of the observation, the average monthly OPD consultation across the various health facilities was 526,881 (a daily average of 17,562.70). There was no significant month-to-month change in the number of consultations before the pandemic as indicated by the p-value of the baseline trend (p = 0.434, p > 0.05). During the Covid-19 pandemic however, we observed a significant decrease in OPD utilization of 151,342.40 (p = 0.03, p < 0.05) month-on-month representing a daily decline of 5,044.75.

Fig 1 presents OPD consultations from January 2018 to December 2021. The vertical dotted line separates the periods precovid-19 (Jan-2019—Feb 2020) and covid-19 (Mar 2020 –Dec 2021). We noticed OPD dropped from 644,163 in Feb 2020 to 586,295 in March 2020 after the confirmed cases of covid-19 in Ghana, representing a sharp decline of 8.98% within the period. This decline continued through to May 2020 before it began to rise again but still below the pre covid-19 period.

Results from the segmented *linear* regression as shown in Table 5 indicates that at the beginning of the period of the observation, the average monthly ANC attendance was 109,795.90 (a daily average of 3,659.86). Before the pandemic, though insignificant, the average

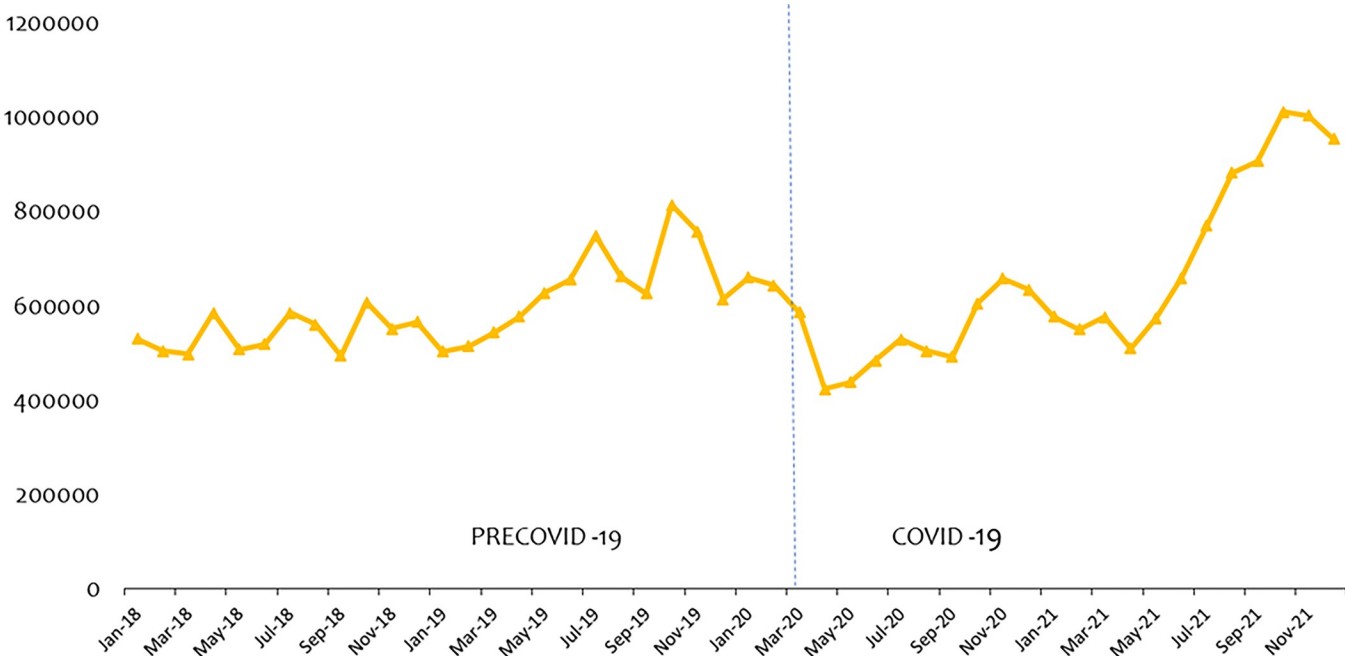

**Fig 1. Trend of outpatient (OPD) consultations.**

**Table 5. Results of segmented linear regression model, ANC attendances.**

| Independent Variables | Coefficient | Std. Error | t | P-value |
|---|---|---|---|---|
| Constant | 109795.9 | 7462.154 | 14.71 | 0.00 |
| Baseline trend | 249.486 | 468.818 | 0.53 | 0.597 |
| Change during covid | -21948.21 | 9311.207 | -2.36 | 0.023 |
| Change in utilization | 3020.756 | 803.682 | 3.76 | 0.00 |

monthly change in ANC attendance was 249.49 (daily average of 8.32 attendance). However, during the pandemic there was a significant monthly decline in ANC attendance of 21,948.21 (p = 0.023, p < 0.05), an estimated daily average of 731.61 reduction in the number of ANC attendance. *Table 6* shows an insignificant monthly increase of 37.76 PNC attendance (daily average of 1.23) compared to monthly PNC attendance of 443.96 before the pandemic. There was also a significant month-to-month change in the number of PNC attendance before the pandemic as indicated by the p-value of the baseline trend (p = 0.00, p < 0.05).

Figs 2 and 3 present the data series of ANC and PNC consultations from January 2018 to December 2021. Fitted with the vertical dotted line to separate the periods labelled precovid-19 and covid-19. We noticed a marginal increase in ANC by 0.99% after the announcement to confirm the covid-19 cases. Interestingly, by Apr 20, ANC attendance had decreased by 11.96% thus from 122,637 Mar 2020 to 107,969 in Apr 2020. Similarly, PNC attendance went up by 7.64% in Mar 2020 and but had decreased by 8.04% in Apr 2020 and a further decline of 2.39% by May 2020.

The segmented linear regression as shown *Table 7* shows an insignificant average monthly increase in delivery case (p = 0.41, p > 0.05) during the pandemic. During the pandemic, there was an average monthly rise of 1,795.83 delivery cases (daily average of 59.86). Nevertheless, this change is higher compared to the insignificant monthly average of 79.76 (p = 0.498, p > 0.05) before the pandemic. The average monthly delivery attendance at the beginning of the observation was 20,744.92 (p = 0.00, p > 0.05) at the beginning of the observation.

Fig 4 shows the longitudinal data of delivery cases recorded from January 2018 to December 2021, fitted with the vertical dotted line to separate the periods labelled precovid-19 and covid-19. As observed from the graph, delivery case rose from 21954 in Feb 2020 to 27,128 in Mar 2020 and then 29,705 by end of May 2020 but started to fall sharply until Aug 2020 (22,778), thus 23.13% fall between May 2020 and Aug 2020.

## 3.1 Did the emergence of covid-19 pandemic lead to a decrease in utilization of services?

As observed in Fig 5, total utilization or hospital attendance after the confirmation of covid-19 in Ghana decreased from 802,100 in Feb 20 to 751,726 in Mar 2020 and continued to decline reaching an all-time lowest hospital attendance of 574,960 by Apr 2020. Although, total attendance had started to rise by May 2020, we observed that the increment was not that significant

**Table 6. Results of segmented linear regression model, PNC attendances.**

| Independent Variables | Coefficient | Std. Error | t | P-value |
|---|---|---|---|---|
| Constant | 2529.639 | 1935.875 | 1.31 | 0.198 |
| Baseline trend | 443.956 | 116.104 | 3.82 | 0.00 |
| Change during covid | 37.763 | 1791.783 | 0.02 | 0.983 |
| Change in utilization | -75.139 | 210.66 | -0.36 | 0.723 |

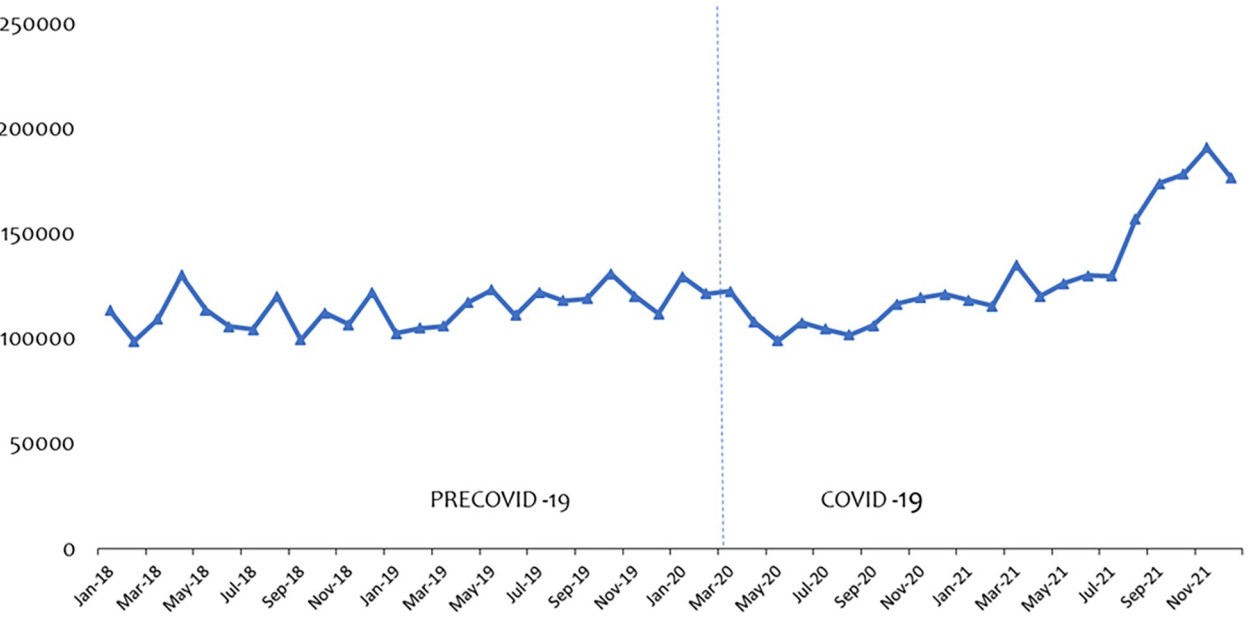

**Fig 2. Trend of ANC attendance over the period.**

compared to the months before the covid-19 until mid-2021 when total attendance started increasing. The commencement and progress of COVID-19 immunization programs might be a possible contributing cause to the observed rise in services from mid-2021. The implementation of vaccination programs on a global scale, including Ghana, is seen as a significant factor in reducing concerns related to visiting healthcare institutions. Vaccination is a fundamental component of public health interventions, serving to regulate the transmission of contagious illnesses. The increasing number of persons who received the COVID-19 vaccination

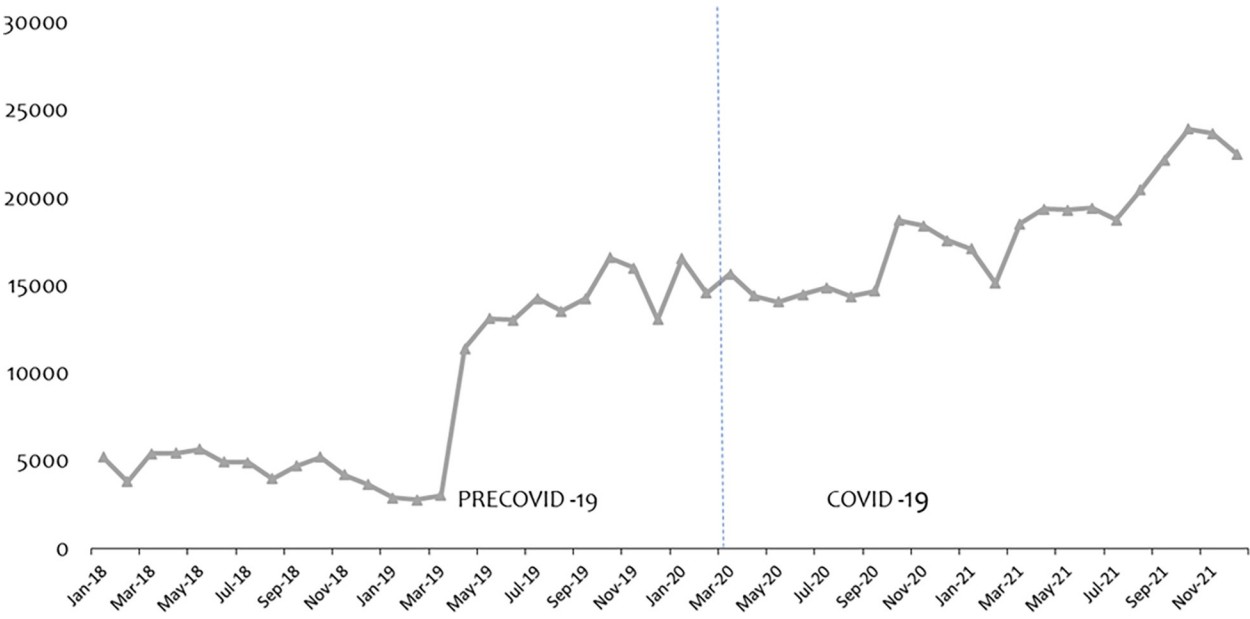

**Fig 3. Trend of PNC attendance over the period.**

**Table 7. Results of segmented linear regression model, DEL attendances.**

| Independent Variables | Coefficient | Std. Error | t | P-value |
|---|---|---|---|---|
| Constant | 20744.92 | 1886.685 | 11.00 | 0.00 |
| Baseline trend | 79.756 | 116.718 | 0.68 | 0.498 |
| Change during covid | 1795.833 | 2158.734 | 0.83 | 0.41 |
| Change in utilization | 319.56 | 204.429 | 1.56 | 0.125 |

likely had a significant impact on boosting confidence among the public. Vaccination has a complex and varied effect on how people seek treatment. As vaccination rates climbed, individuals who were previously reluctant to visit healthcare facilities owing to infection fears may have felt more confident, which influenced their decision-making regarding their health. The vaccine's function in alleviating severe sickness and diminishing transmission may have played a part in a gradual resumption of healthcare-seeking habits. Furthermore, the relaxation of regulations, together with increased public awareness and education, may have synergistically contributed to the noticeable increase in attendance. The gradual resumption of normalcy and the alleviation of concerns associated to the epidemic certainly motivated individuals to begin seeking healthcare services.

## 3.2 The way in which utilization was impacted differ across the different levels of care

Claims data from across seven (7) different health facility types/levels were collected for the study as detailed in *Tables* 1 and 4 in S1 Appendix. Fig 6 depicts total hospital attendance across these different facility levels. Hospital attendance in tertiary hospital dropped from 23,453 in Feb 20 to 21,446 in Mar 20; secondary hospital from 94,956 to 88,641; primary hospitals from 501,650 to 464,572; health centres from 120,372 to 109,219; clinics from 39,429 to 36,812; maternity home from 7,450 to 5,540. Contrastingly, utilization in CHPS compounds

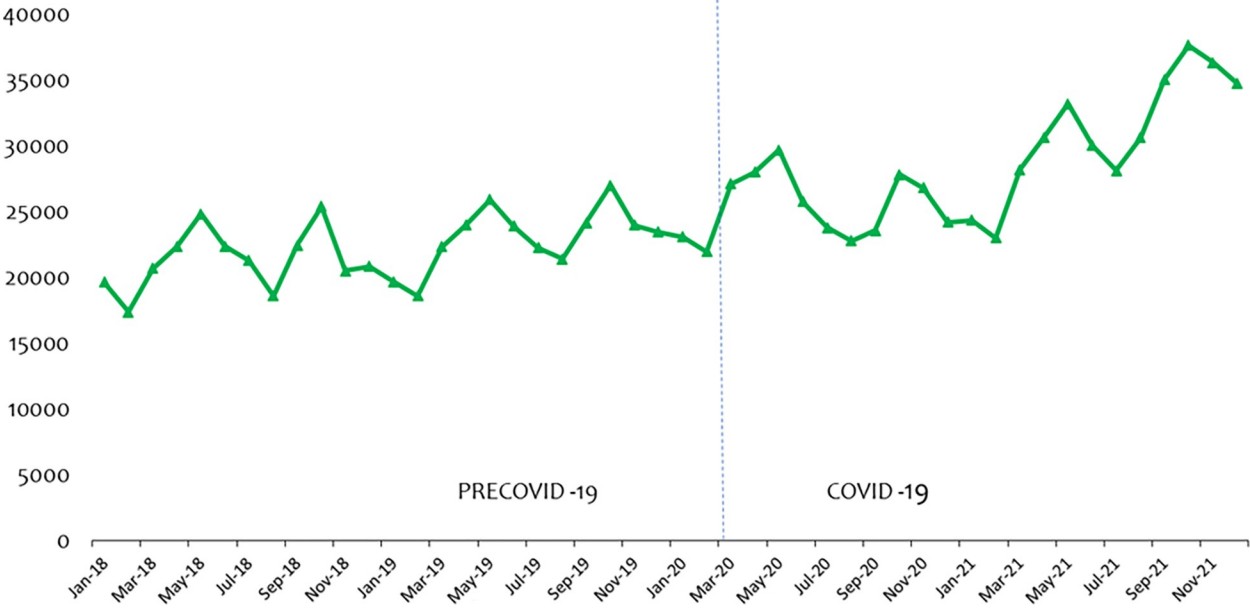

**Fig 4. Trend of delivery (DEL) consultations over the period.**

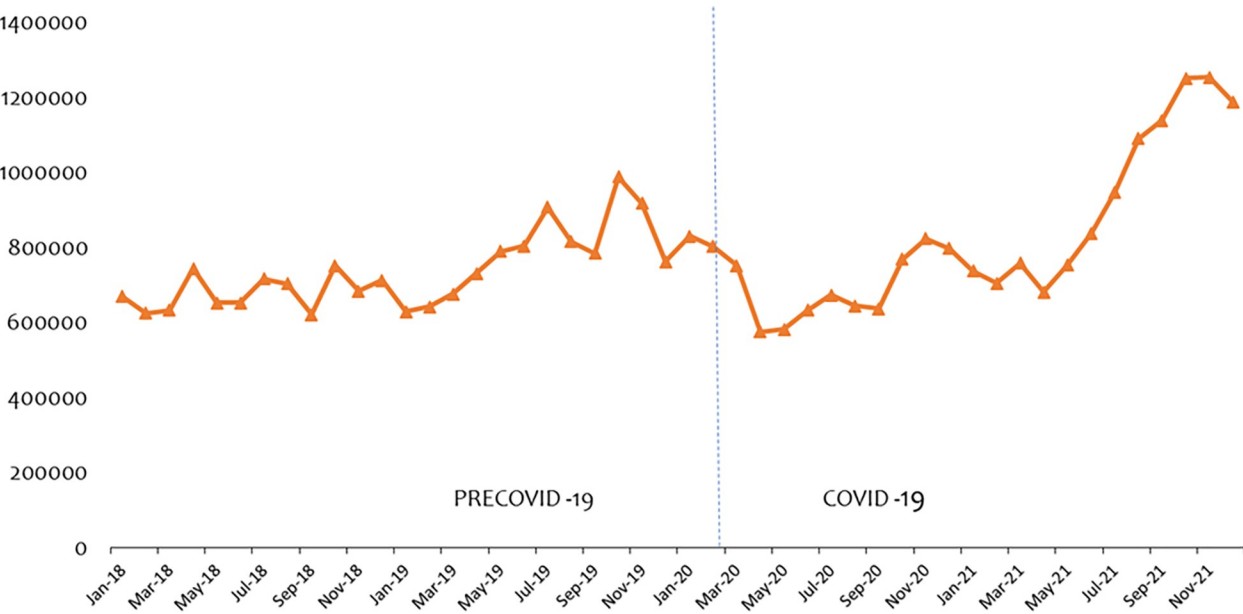

**Fig 5. Trend of total hospital attendance.**

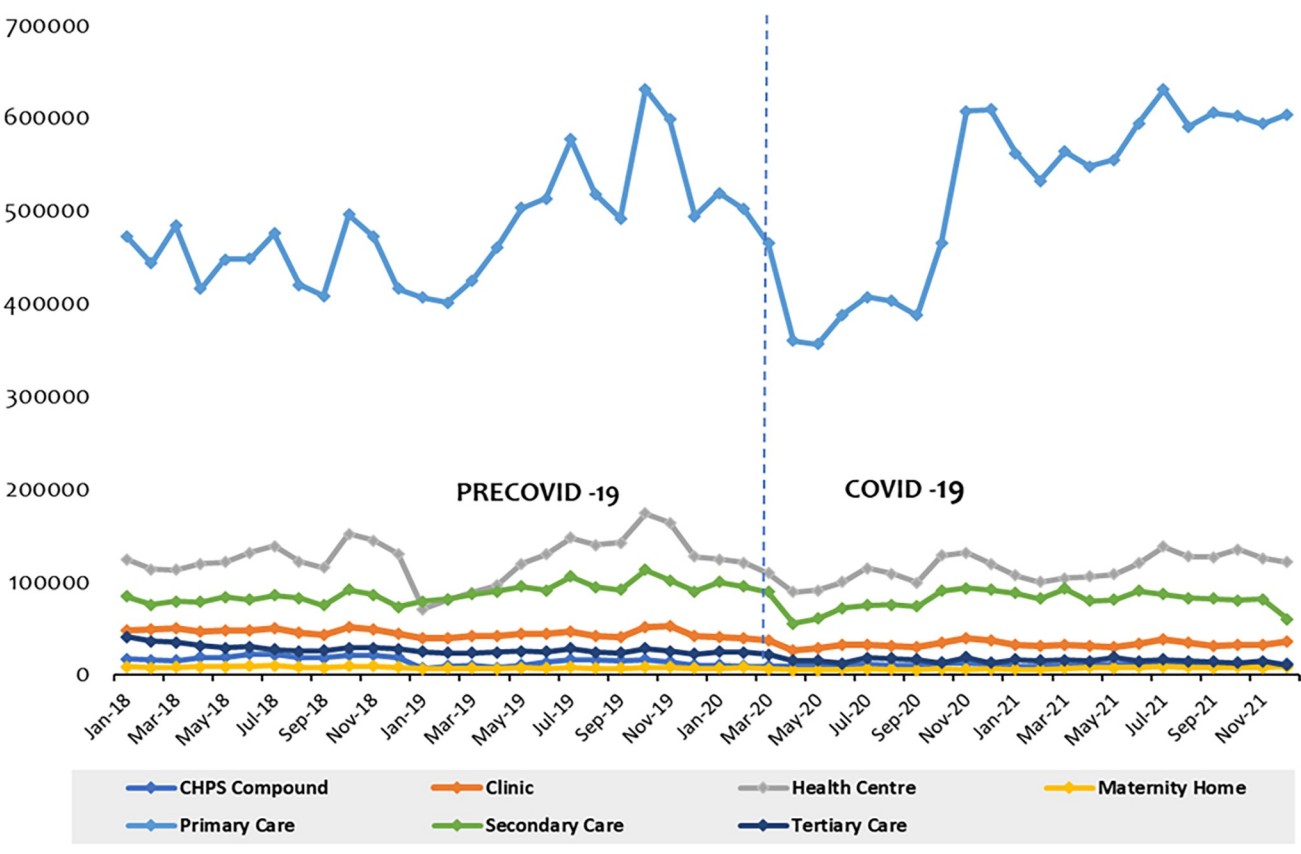

**Fig 6. Trend of total hospital attendance by facility type.**

increase from 8,888 in Feb-20 to 8,905 in Mar-20. A month-on-month dynamic of hospital attendance across these facility levels is shown in *Table* 4 in S1 Appendix. Also, the average monthly hospital attendance across these facility levels for the periods precovid-19 and covid-19 are shown in *Table 5* in S1 Appendix. Fig 2 in S2 Appendix shows the total utilization for each specialty of interest by facility type/level.

## 4. Discussion

The study's findings indicate that there was a significant decline in the volume of OPD health-care utilization at all the various levels of healthcare facilities in Ghana compared to the pre covid-19 period. This decrease is consistent with Andersen's *Behavioral Model of Health Services Utilization* [17]. Andersen's model, a widely recognized model, suggests that healthcare utilization is influenced by a combination of predisposing factors (such as individual traits and beliefs), enabling factors (related to healthcare resource accessibility), and need factors (associated with perceived health condition). Regarding the observed decrease in OPD usage, the variables that may contribute to this fall include individual views and beliefs about the likelihood of getting COVID-19 in healthcare settings. The apprehension of being exposed to the virus may serve as a barrier, prompting persons to abstain from receiving regular outpatient care. Enabling factors, which are variables that either enable or impede access to healthcare, may encompass limitations on mobility and difficulties in transportation due to the pandemic. These constraints reduce persons' capacity to attend healthcare facilities for outpatient appointments. In addition, the influence of necessary components, which encompass an individual's subjective assessment of their health condition, is of utmost importance. The modified health requirements during the pandemic, along with the emphasis on COVID-19-related issues, may have led to a decline in the perceived necessity for regular outpatient department (OPD) treatments. The extent of this fall is emphasized by a monthly reduction of more than 5000 daily OPD attendances. The data also shows that there was a 28% decrease in OPD attendance a month after the covid-19 infection was announced and restriction in movements enforced in Ghana. This outcome is comparable to other studies carried in other parts of the world. A review of literature on covid-19 conducted by Roy et al. [5] indicated that healthcare utilization decreased universally across both lower and high income countries. Similarly, a study in Sierra Leone by Sevalie et al. [18] also concluded that, there was a significant decrease in hospital utilizations after the first reported case of covid-19 and this trend continued to the third quarter of 2020 as compared with data during the precovid-19 pandemic.

Relative to maternal healthcare, the data clearly shows there was a significant drop in the number of expectant mothers visiting the health facility for ANC visits. This decrease is in keeping with the principles of the *Health Belief Model* (HBM) [19]. According to the HBM, individuals are less likely to engage in preventive health practices if they believe they have a reduced likelihood of experiencing a health danger. Amidst the Covid-19 outbreak, the noticeable decrease in ANC visits signifies a complex interaction between individuals' perceived vulnerability to the virus and their inclination to seek healthcare. The HBM also suggests that individuals assess perceived advantages in comparison to perceived obstacles when making decisions connected to their health. For pregnant women during the pandemic, worries about being exposed to the virus at healthcare facilities may have been seen as a significant obstacle, which seemed more important than the advantages of attending antenatal care sessions. The complex relationship between how likely people think they are to be affected by a health issue, how serious they think the consequences of that issue are, and the overall evaluation of the advantages and obstacles involved, helps us understand the many aspects involved in making decisions about preventive healthcare activities. Furthermore, the HBM highlights the

importance of signals to action, highlighting the relevance of external stimuli that motivate individuals to participate in activities that promote their health. The extensive circulation of information on the dangers and preventive measures during the pandemic acted as a crucial stimulus for action, which had an impact on the reported decrease of ANC visits. The apprehension of acquiring the virus, along with modified perceptions of the imminent health risk, probably had a role in the significant change in healthcare-seeking conduct among pregnant women. Our findings are also corroborated by Almas, Afsheen, Memon, & Avesi [20] who reported in their study on Ethiopia that there was a significant drop in pregnant women attending ANC during the covid-19 pandemic. In this study, there is a monthly decline of more than 20,000 attendees (p < 0.05). Similar to the ANC data, PNC also saw a decline in its attendance after the announcement of detection of covid-19 in Ghana compared to the pre covid-19 period. There was an increase in PNC attendance for the first month of the announcement though and this could be due to mothers who had just delivered using their new born babies as a reason to visit the PNC clinic for their personal healthcare concerns since the general OPD had restrictions due to the pandemic. However, contrary to the ANC, the decline in attendance for the PNC was not statistically significant *(p = 0.98)*.

The data did not show any significant change in deliveries of pregnant women as a result of the covid-19 pandemic. This study did not find any comparable literature to buttress or otherwise this finding. However, it was observed that the number of deliveries rose significantly a year after the restriction of movement of persons. These restrictions kept people indoors which probably gave couples enough time to be together and this might have resulted in increased reproduction. The study had envisaged a decline in the number of deliveries at health facilities due to the pandemic. However, this was not the case but it cannot be concluded that home deliveries did not occur during the pandemic period as this study did not include that. This intriguing trend aligns with the *Diffusion of Innovations theory* [21], which posits that the adoption of new behaviors is influenced by communication channels and perceived benefits of innovation. The significant upswing in deliveries a year later suggests that information dissemination channels regarding childbirth, coupled with the adoption of innovative practices and trust in the healthcare system, may have continued to exert influence. Furthermore, it is plausible that women, despite the challenges posed by the pandemic, chose to stick with hospitals for deliveries due to the perceived clear benefits compared to conventional delivery methods. This underscores the enduring trust in healthcare facilities and their perceived advantages, even amid unprecedented circumstances. Finally, the data shows that, total utilization after the pandemic was announced in Ghana saw a decline in all the different facility levels of care. CHPS facilities were the only ones that saw no decline but rather a slight increase. This is attributable to the fact that CHPS facilities are community-based and the people see it to be within their reach and also could be free from the pandemic since it is within their community. Moreover, since the CHPS serve a relatively small community, overcrowding is less likely to occur. Hence, issues such as limitations on the number of people and long waiting hours occasioned by the restrictions were practically non-existent. People were, therefore not deterred from obtaining healthcare from CHPS facilities.

The decrease in utilization is due to a couple of factors. The conception that the hospitals can be a source of infection spread as the patients of the covid-19 infection are being treated there could be one of the reasons. In Ethiopia, Almas et al., [17] showed that 87% of pregnant women did not attend ANC for the fear of contracting the infection. This conception prevented people who needed healthcare from visiting the healthcare facility. Another contributory factor could be the restriction in movement of people in the early months of the pandemic in Ghana. These restrictions affected some health facilities because they could not operate to their full capacity because though health staffs were exempted from the movement

restrictions, commuting to work by the staff was a challenge as commercial vehicle drivers who would transport them were restricted in movement. Lastly, social distancing during the pandemic may have also contributed to the decline. The spacing meant a limited number of patients can be allowed at the OPD for consultation. Also, a lot of time was involved in going through the health and safety protocols before being attended to. The stress of going through these processes may have contributed to the people not visiting the healthcare facilities and hence the decline. Healthcare workers, being front liners easily contracted the virus and had to isolate as per protocol decreasing the already low numbers of healthcare personnel to cater for health needs.

This reduction in healthcare utilization as evidently shown in this study indicates that there is an unmet need in the health system that needs to be addressed. Preparedness of the Ghana health system in times of pandemics is weak and it is compelling to make efforts to prioritise the need for solutions to reduce or prevent missed healthcare in times like this. The need to leverage on technology to deliver healthcare in Ghana is more pronounced than ever. Introduction of telemedicine to facilitate healthcare delivery in hard-to-reach areas where there are no consultants or healthcare delivery personnel during pandemic like this should be enhanced. Electronic-Pharmacy (e-pharmacy) can also be introduced to help serve chronic patients who are on long term routine medications. These patients can either pick up their medications from the pharmacies or delivered to them at their homes through couriers without them necessarily visiting the hospital where they will be worried of being infected with the virus.

## 5. Conclusion

This study set out to determine the effect of covid-19 pandemic on healthcare utilization in general with emphasis on pregnant women and newborn care in Ghana. Our findings have serious policy implications on the health system's preparedness for healthcare during furture pandemics. The study revealed that there was a decline in healthcare utilization in Ghana with the onset of Covid-19 pandemic. Overall, OPD, ANC and PNC attendance saw a decline in its utilization which is comparable to other studies around the world. There was no significant change in the number of newborn babies delivered during the period of study which may need further studies to determine the number of baby deliveries that took place outside the healthcare facility during the period.

The results from CHPS facilities demonstrate the resilience of the concept of community healthcare delivery. It proves that the community health delivery system is effective and efficient and should be enhanced across the country to improve on the healthcare delivery to help achieve the Universal Health Coverage (UHC) by year 2030. The study also shows that there is an unmet need in the health delivery system of the country during times of difficulty such as pandemics and natural disasters. The inclusion of technology such as Tele-health and e-pharmacy in the health delivery will greatly improve on access to quality healthcare in times of difficulty. The distribution of the limited resources available to areas of priority has been showcased in this study and efforts should be made by policy makers to appropriately allocate the resources to ensure an improved healthcare for the people.

## Supporting information

**S1 Appendix.**
(PDF)

**S2 Appendix.**
(PDF)

**S3 Appendix.**
(DOCX)

## Acknowledgments

We thank the staff at the four Claims Processing Centres (CPCs) at the National Health Insurance Authority for facilitating access to the data used for the analysis.

## Author Contributions

**Conceptualization:** Yaw Nyarko Opoku-Boateng, Mylene Lagarde.

**Data curation:** Yaw Nyarko Opoku-Boateng, Emmanuel Opoku-Asante.

**Formal analysis:** Yaw Nyarko Opoku-Boateng, Emmanuel Opoku-Asante.

**Funding acquisition:** Yaw Nyarko Opoku-Boateng.

**Investigation:** Yaw Nyarko Opoku-Boateng, Emmanuel Opoku-Asante, Edward Nketiah-Amponsah.

**Methodology:** Yaw Nyarko Opoku-Boateng, Emmanuel Opoku-Asante, Mylene Lagarde, Edward Nketiah-Amponsah.

**Project administration:** Yaw Nyarko Opoku-Boateng.

**Resources:** Yaw Nyarko Opoku-Boateng, Emmanuel Opoku-Asante, Mylene Lagarde, Edward Nketiah-Amponsah.

**Supervision:** Mylene Lagarde.

**Validation:** Yaw Nyarko Opoku-Boateng, Emmanuel Opoku-Asante, Mylene Lagarde, Edward Nketiah-Amponsah.

**Visualization:** Yaw Nyarko Opoku-Boateng, Emmanuel Opoku-Asante, Mylene Lagarde, Edward Nketiah-Amponsah.

**Writing – original draft:** Yaw Nyarko Opoku-Boateng.

**Writing – review & editing:** Yaw Nyarko Opoku-Boateng, Emmanuel Opoku-Asante, Mylene Lagarde, Edward Nketiah-Amponsah.

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
