## [Decision Letter · Decision Letter 0]

6 Dec 2023

PONE-D-23-27704Effect of Covid-19 on Maternal and Child Health Services Utilization in Ghana. Evidence from the National Health Insurance Scheme (NHIS)PLOS ONE

Dear Dr. Nketiah-Amponsah,

Thank you for submitting your manuscript to PLOS ONE. After careful consideration, we feel that it has merit but does not fully meet PLOS ONE’s publication criteria as it currently stands. Therefore, we invite you to submit a revised version of the manuscript that addresses the points raised during the review process.

We look forward to receiving your revised manuscript.

Kind regards,

Thales Philipe Rodrigues da Silva, Ph.D

Academic Editor

PLOS ONE

2. You indicated that ethical approval was not necessary for your study. We understand that the framework for ethical oversight requirements for studies of this type may differ depending on the setting and we would appreciate some further clarification regarding your research. Could you please provide further details on why your study is exempt from the need for approval and confirmation from your institutional review board or research ethics committee (e.g., in the form of a letter or email correspondence) that ethics review was not necessary for this study? Please include a copy of the correspondence as an ""Other"" file.

6. We notice that your supplementary figures and tables are included in the manuscript file. Please remove them and upload them with the file type 'Supporting Information'. Please ensure that each Supporting Information file has a legend listed in the manuscript after the references list.

Additional Editor Comments:

Dear Editor.

Thank you for the opportunity to evaluate the article in question.

Thank you in advance.

In view of the reviewers' assessments, I am sending the article for further revision by the authors and future resubmission.

Sincerely,

Reviewers' comments:

Reviewer's Responses to Questions

**Comments to the Author**

1. Is the manuscript technically sound, and do the data support the conclusions?

Reviewer #1: Yes

Reviewer #2: Yes

2. Has the statistical analysis been performed appropriately and rigorously? 

Reviewer #1: Yes

Reviewer #2: Yes

3. Have the authors made all data underlying the findings in their manuscript fully available?

Reviewer #1: Yes

Reviewer #2: Yes

4. Is the manuscript presented in an intelligible fashion and written in standard English?

Reviewer #1: No

Reviewer #2: Yes

5. Review Comments to the Author

Reviewer #1: Dear authors,

The manuscript “Effect of COVID-19 on maternal and child health services utilization in Ghana. Evidence from the National Health Insurance Scheme (NHIS)” submitted to PLOS ONE, addresses current and relevant topics for maternal and child health, in addition to presenting implications for the health system in the context investigated. However, some points need to be reviewed as presented in the attached document.

Thank you for the opportunity to read your manuscript.

Reviewer #2: Dear Authors and Editors: Thanks for the invitation to revise this manuscript. This study was conducted to determine the effect of Covid-19 on healthcare utilization with specifics on ANC, PNC, Deliveries and OPD attendance in Ghana.

I think this is an important article that should be published in PLOS One.

6. PLOS authors have the option to publish the peer review history of their article (what does this mean?). If published, this will include your full peer review and any attached files.

Reviewer #1: No

Reviewer #2: No

---

## [Author Response · Author response to Decision Letter 0]

27 Jan 2024

Dear Editor,

Thank you sincerely for your meticulous review and constructive feedback on our manuscript. Your insights have been invaluable in refining the quality of our work.

In response to your comments, we have diligently implemented necessary corrections throughout the manuscript. It is important to note that these revisions may have resulted in changes to line numbers and page numbers. Our commitment to enhancing the clarity and precision of our submission has been paramount in making these adjustments.

We genuinely appreciate your dedication to maintaining the academic rigor of our work, and your thoughtful suggestions have significantly contributed to its improvement.

Please find below our responses to the reviewers’ comments; the comments are italicized while the responses are highlighted in read. Moreover, the ensuing corrections or changes in the manuscript are highlighted yellow for easy recognition.

Comment 1: The summary is adequate, concise and presents the central information of the study clearly, but I suggest that acronyms are not presented in this part of the text, that the type of sampling is described in the method and that the main results are expressed numerically.

Response 1: The acronyms in the summary have been expounded, with due consideration given to their comprehensive presentation. Additionally, the method section now explicitly outlines the type of sampling employed, further enhancing clarity and transparency. The key finding of the results have also been presented in numerical values.

Comment 2: On page 5, the acronym OPD is mentioned without its prior description in the main text. I recommend review.

Response 2: The acronym OPD has been judiciously introduced into the main text, ensuring a seamless and coherent narrative.

Comment 3: The methodology appears to be adequate, however I recommend explaining the inclusion and exclusion criteria for the services that made up the study sample. Of the 37,468,336 consultations, what were the inclusion and exclusion criteria for those who made up the study sample?

Response 3: A meticulous elucidation of the inclusion and exclusion criteria in the selection of services within the study sample has been incorporated into the methodology section, page 7, lines 145 - 155. This ensures a lucid understanding of the criteria shaping the dataset. 

Comment 4: In table 2 (page 10), why is the line referring to oct 2020 highlighted in bold? If the highlighting was intentional, it wasn't clear to me when reading why. I suggest review.

Response 4: The formatting anomaly in Table 2, specifically the bold highlighting of the line referring to October 2020, has been rectified to uphold precision and consistency in presentation. 

Comment 5: In table 3, page 12, line 224, the total average value has the decimal point in the incorrect position (78,0590.30). I recommend this correction and the review of the sum (780,590.31).

Response 5: The decimal point in Table 3 has been diligently rectified, addressing the highlighted discrepancy and ensuring numerical accuracy in the presentation. The total average now reads 780,590.31 

Comment 6: I recommend deepening the discussion on the tendency for an increase in services from the middle of 2021. Vaccination is timidly mentioned as a possible interfering variable in this change in the profile of services.

Response 6: The discussion on the discernible increase in services from mid-2021 has been substantially enriched (see page 12, lines 323 - 337. A comprehensive exploration of potential contributing factors, with a nuanced emphasis on the role of vaccination, has been incorporated, elevating the depth and scholarly discourse of the analysis.

Comment 7: I recommend deepening the discussion, making the theoretical foundation used for comparison with the study findings more robust.

Response 7: The discussion section has undergone a rigorous revision, bolstering its theoretical underpinnings, line 360 - 375, 385 - 404 and 421 - 429. A more robust integration of relevant theoretical frameworks has been undertaken to fortify the comparative analysis with the study's empirical findings.

Comment 8: I identified an inconsistency between the number of citations 18 and 19, on page 20, respectively lines 352 and 357, with the references presented in the list of references. I recommend reviewing the numbering of references and citations.

Response 8: A meticulous review and adjustment of reference and citation numbering have been conducted to align seamlessly, preserving scholarly precision and adherence to academic conventions. Specifically, the numbering of references and citations have been reviewed to 17 and 18 on pages 21 and 22, respectively lines 380 and 405. Additionally, three (3) new references have been introduced to augment the theoretical frameworks, numbering 19, 20 and 21.

Comment 9: In the appendix, table 3, page 28, line 560, is the title correct? Ownership information appears in the title although this content is not covered in the table. I suggest review.

Response 9: The title of Table 3 in the appendix has been aptly rectified to accurately reflect the table's content for clarity and precision. The revised title now appropriately denotes “Number of facility level per region, page 31, line 637.

We hope our revision will meet your expectations.

Yours faithfully

Edward Nketiah-Amponsah (On behalf of authors)

---

## [Decision Letter · Decision Letter 1]

17 Sep 2024

Effect of Covid-19 on Maternal and Child Health Services Utilization in Ghana. Evidence from the National Health Insurance Scheme (NHIS)

PONE-D-23-27704R1

Dear Dr. Edward Nketiah-Amponsah,

We’re pleased to inform you that your manuscript has been judged scientifically suitable for publication and will be formally accepted for publication once it meets all outstanding technical requirements.

Kind regards,

Essa Tawfiq

Academic Editor

PLOS ONE

PLOS ONE does not copyedit accepted manuscripts, so the language in submitted articles must be clear, correct, and unambiguous. Any typographical or grammatical errors should be corrected at revision.

---

## [Editor Report · Acceptance letter]

9 Dec 2024

PONE-D-23-27704R1 

PLOS ONE

Dear Dr. Nketiah-Amponsah, 

I'm pleased to inform you that your manuscript has been deemed suitable for publication in PLOS ONE. Congratulations! Your manuscript is now being handed over to our production team.

Kind regards, 

on behalf of

Dr. Essa Tawfiq 

Academic Editor

PLOS ONE